# A Diffusion-based Generative Approach for Model-free Finite-time Control of Complex Systems

## Abstract

Complex systems with nonlinear dynamics pose significant challenges for finite-time optimal control, especially when accurate system models are unavailable. This paper introduces DIFOCON (DIffusion Finite-time Optimal CONtrol), a novel data-driven framework for finite-time optimal control that operates without prior knowledge of system parameters or dynamics. DIFOCON reformulates the control problem as a generative task, optimizing control signal trajectories to guide systems to target states within a finite time. Our approach utilizes a diffusion model with a dual-Unet architecture to capture nonlinear system dynamics and generate entire control sequences in a single step. Additionally, an inverse dynamics module is integrated to ensure that the generated control signals are appropriate for complex systems. To further enhance performance, we propose a retraining strategy that improves out-of-distribution generalization. Experiments on two nonlinear complex systems demonstrate DIFOCON's superior performance, reducing target loss by over 26.9% and control energy by over 15.8% compared to baselines while achieving up to 4 times faster convergence in practical steering tasks. The implementation of this work can be found at https://anonymous.4open.science/r/DIFOCON-C019/.

## 1 Introduction

Complex systems are composed of interacting components that exhibit emergent behaviors and nonlinear dynamics (Ladyman et al., 2013). Control problems are fundamental in both natural and engineering systems, where the goal is to direct system behavior toward desired outcomes by determining appropriate inputs. *Finite-time control*, which requires systems to reach objectives within a limited time, is particularly critical but challenging, especially when accurate system models are unavailable.

Existing control methodologies predominantly rely on precise system models to achieve desired outcomes (Yan et al., 2012; Lindmark & Altafini, 2018; Liu et al., 2011; Gao et al., 2014). However, in real-world applications, obtaining accurate system equations and parameters is often impractical, especially for complex systems with nonlinear dynamics. While a few efforts toward model-free approaches have been made, significant challenges remain. Feedback-based control methods, such as PID and reinforcement learning (Li et al., 2006; Hwang et al., 2022; Pomerleau, 1988; Haarnoja et al., 2018), rely on continuous cycles of sensing, processing, and decision-making, which makes them computationally intensive and often unsuitable for high-dimensional, finite-time control problems. Baggio et al. (2021) developed an optimal control solution for linear systems by implicitly estimating system equations and parameters from empirical data, but their approach struggles when applied to nonlinear systems.

In this paper, we propose a novel framework for finite-time optimal control, DIffusion Finite-time Optimal CONtrol (DIFOCON), which operates solely based on data, without requiring prior knowledge of system parameters or dynamics. By reframing the control problem as a generative task, our approach seeks to optimize the trajectory of control signals that guide the system to its target state within a finite time. Through data-driven generative modeling, DIFOCON learns the distribution of control signals and observed states, generating complete control sequences in a single step, thus

eliminating the need for continuous feedback-based interaction with the system. This makes our approach highly efficient and well-suited for high-dimensional, nonlinear systems.

The core of DIFOCON is built on a diffusion model (Ho et al., 2020) that progressively refines noisy inputs into high-quality control sequences conditioned on desired system outcomes. To capture non-linear dynamics, we introduce a dual-Unet architecture with residual connections in the denoising network, which accounts for both first-order and higher-order expansions, based on optimal control theory initially developed for linear systems. To ensure the generated signals are well-suited for complex systems with unknown dynamics, we integrate an inverse dynamics module, which computes control signals from denoised states, rather than directly sampling both control signals and intermediate states using the denoising network. Additionally, to enhance the model's ability to generate out-of-distribution data from the training data, we propose a retraining method that fine-tunes the model on its own generated samples, thereby expanding the exploration space and improving generalization.

Our experiments on two typical nonlinear complex systems (Acebrón et al., 2005; Susuki et al., 2011) demonstrate DIFOCON's superior performance in solving finite-time optimal control problems. Specifically, we show that DIFOCON achieves the best in-distribution performance, reducing target loss by over 26.9% and control energy by 15.8% compared to the best-performing baseline when applied to systems governed by first-order and second-order ordinary differential equations. Moreover, our method exhibits, at most, around 4 times faster convergence in practical steering tasks compared to existing model-free approaches. These results validate the efficacy of our diffusion-based generative model in addressing the challenges of finite-time control in complex, real-world systems.

## 2 RELATED WORKS

### 2.1 MODEL-FREE CONTROL OF COMPLEX SYSTEMS

Model-free control approaches circumvent the need for an exact model by leveraging control data to capture the system's dynamics implicitly. These methods can be categorized into closed-loop and finite-time control strategies.

**Closed-loop control methods.** Classical closed-loop control methods like Proportional-Integral-Derivative (PID) (Li et al., 2006) are famous for their steadiness and efficiency but face challenges in adaptability in high-dimensional complex scenarios. In the realm of deep learning, reinforcement learning (Pomerleau, 1988; Haarnoja et al., 2018) has recently demonstrated its effectiveness in sequential decision making, and supervised learning methods (Hwang et al., 2022) show their adaptability by using neural surrogate models to learn control sequences. However, the above closed-loop control methods are predominantly employed for stabilization or tracking tasks, yet they fall short when applied to real-world scenarios where control operations are subject to time constraints.

**Finite-time control methods.** Finite time control methods (Baggio et al., 2021; Wei et al., 2024), on the other hand, optimize the control sequence over the entire horizon, thus addressing the myopic nature of closed-loop approaches and are suitable for tasks that require control of system states within a finite time. Notably, Baggio et al. (2021) proposed an analytical method that leverages data to determine the optimal input for steady-state control of complex networks, without knowing the dynamics. However, its foundation in linear systems theory restricts its generalization to nonlinear dynamics, impeding its ability to accurately steer the system toward the desired state.

### 2.2 DIFFUSION MODEL

Denoising diffusion probabilistic models (Ho et al., 2020) have gathered significant attention for their ability to generate high-quality and consistent samples across various domains such as image, audio, and video, achieving state-of-the-art (SOTA) results (Dhariwal & Nichol, 2021; Kong et al., 2020; Ho et al., 2022). These models have also demonstrated their capability in traditional mathematical and engineering problems, including optimization (Krishnamoorthy et al., 2023; Sun & Yang, 2023), inverse problems (Chung et al., 2022), robotic control (Janner et al., 2022; Ajay et al., 2022), etc. Wei et al. (2024) introduced DiffPhyCon, a finite time control method that harnesses generative diffusion models to directly optimize system trajectories and control sequences

over the entire horizon. This approach implicitly captures the inherent constraints within the system dynamics and employs reweighting techniques during the sampling process to guide the generation of optimal trajectories that deviate from the distribution. However, by implicitly capturing dynamics across the entire trajectory, this method overlooks the Markovian nature of time-invariant systems, and the control sequences are often non-smooth, making it challenging for diffusion models to model their distribution accurately. Our proposed method employs a parameterized inverse dynamics model to explicitly model the relationship between actions and states, allowing the generated control sequences to more accurately guide the evolution of system states.

## 3 BACKGROUND

### 3.1 PROBLEM FORMULATION OF MODEL-FREE FINITE TIME OPTIMAL CONTROL

The dynamics of a controlled system can be written as $\dot{\mathbf{y}}_t = D(\mathbf{y}_t, \mathbf{u}_t)$, where $\mathbf{y}_t \in \mathbb{R}^N$ is the observed system state and $\mathbf{u}_t \in \mathbb{R}^M$ is the control function. Assuming that the system is controllable in a finite time $T$, that is, the system state can be controlled to a desired state $\mathbf{y}_f$ starting from $\mathbf{y}_0$. In most cases, there is also a need to optimize the cost of energy from the control input, i.e., $J(\mathbf{u}) = \int_0^T |\mathbf{u}(t')|^2 dt'$. The formulation **of Model-free (data-driven) Optimal control** is as follows: when the non-optimal control trajectory dataset $D = \{\mathbf{u}^{(i)}, \mathbf{y}^{(i)}\}$, $i \in \{1, 2, \ldots, P\}$ ($P$ is the number of data) is collected from observation of the system, the goal is to find the control signal that minimizes the cost function and the final distance to target state:

$$\mathbf{u}^* = \arg\min_{\mathbf{u}}(J(\mathbf{u}) + L(\mathbf{y}(T), \mathbf{y}_f)) \quad \text{subject to} \quad \Psi(\mathbf{u}, \mathbf{y}) = 0 \tag{1}$$

where $L(\mathbf{y}_T, \mathbf{y}_f)$ denote the distance between the target and final state, and $\Psi(\mathbf{u}, \mathbf{y}) = 0$ is the system dynamic constraint that explicitly provides the starting state $\mathbf{y}_0$ and is implicitly specified by the dataset $D$. The complex dynamics of systems pose a great challenge for solving the problem in this setting.

### 3.2 DIFFUSION MODELS

Diffusion models have emerged as a powerful class of generative models in recent years, demonstrating remarkable capabilities in image synthesis, audio generation, and other domains (Ho et al., 2020; Dhariwal & Nichol, 2021). These models are based on the principle of gradually adding noise to data and then learning to reverse this process. In the forward process, a data point $x^0$ is progressively corrupted through $T$ timesteps, resulting in a sequence of increasingly noisy versions $x^1, x^2, ..., x^K$. Each step applies a small amount of Gaussian noise: $q(x^k|x^{k-1}) = \mathcal{N}(x^k; \sqrt{1-\beta^k}x^{k-1}, \beta^k\mathbf{I})$,

where $\beta^k$ is a variance schedule that controls the noise level. In the reverse process, the model learns to gradually denoise the data, starting from pure noise $x^K$ and working backward to reconstruct the original data point $x^0$. This is modeled as: $p_\theta(x^{k-1}|x^k) = \mathcal{N}(x^{k-1}; \mu_\theta(x^k, k), \Sigma_\theta(x^k, k))$, where $\theta$ represents the learnable parameters of the model. The training objective for diffusion models typically involves minimizing the variational lower bound (VLB) on the negative log-likelihood (Sohl-Dickstein et al., 2015). In practice, this often reduces to a form of denoising score matching (Song & Ermon, 2019).

### 3.3 CLASSIFIER-FREE GUIDANCE

Classifier-free guidance is a technique introduced by Ho & Salimans (2022) to enhance the sample quality and controllability of diffusion models without requiring a separate classifier. During training, randomly set the conditioning $y$ to a null token (e.g., empty string or zero vector) with probability $p$. This allows the model to learn both conditional and unconditional generation. The model is trained using a weighted combination of conditional and unconditional losses:

$$\mathcal{L} = \mathbb{E}_{x^0, \epsilon, k, y}\left[(1-p)\|\epsilon - \epsilon_\theta(x^k, t, y)\|^2 + p\|\epsilon - \epsilon_\theta(x^k, k, \emptyset)\|^2\right] \tag{2}$$

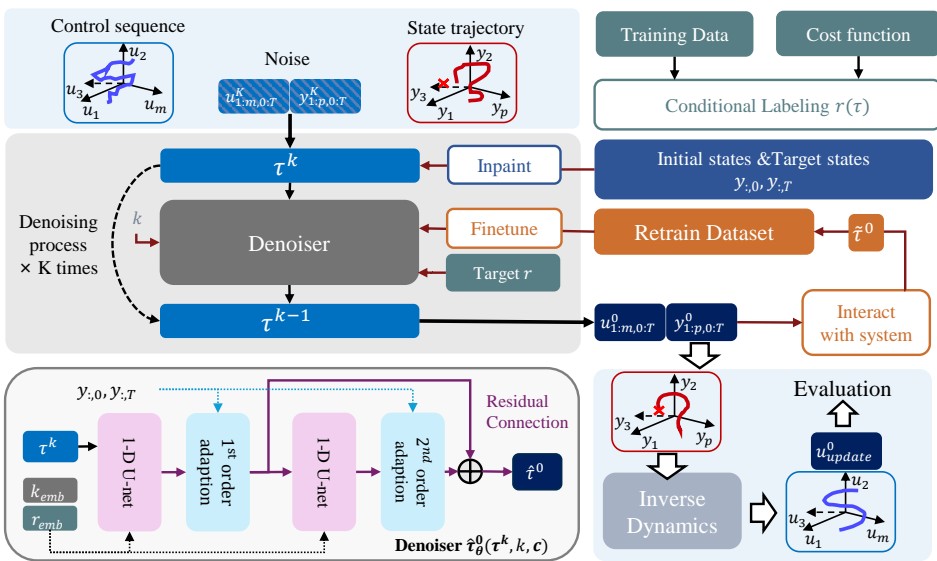

Figure 1: Illustration of the proposed conditional diffusion model-based controller.

where $\epsilon$ is the noise added to the clean data $x^0$ to obtain $x^k$, and $\|\cdot\|$ denotes the L2 norm. At inference time, it introduces a guidance scale $w$ and sample using:

$$\tilde{\epsilon}_\theta(x^k, k, y) = (1+w)\epsilon_\theta(x^k, k, y) - w\epsilon_\theta(x^k, k, \emptyset) \tag{3}$$

where $\tilde{\epsilon}_\theta$ is the guided noise prediction. This formulation allows for controlled generation without needing a separate classifier, offering a more streamlined and efficient method for guided synthesis. The guidance scale $w$ controls the trade-off between sample quality and adherence to the conditioning information. Higher values of $w$ typically result in samples that more closely match the conditioning but may sacrifice some diversity or realism.

## 4 METHOD

### 4.1 CONTROLLING USING CONDITIONAL DIFFUSION MODEL

We formulate the control problem as a conditional generative modeling task. Providing control data $\boldsymbol{\tau} := [\mathbf{u}, \mathbf{y}]$, which is composed of control sequences $\mathbf{u} \in \mathbb{R}^{T \times M}$ and corresponding observation states $\mathbf{y} \in \mathbb{R}^{T \times N}$ of the system, the objective of finding optimal control can be formulated as conditional generative modeling where we obtain a distribution $p_\theta$ to generate high-likelihood control given target and optimization goal:

$$\max_\theta \mathbb{E}_{\boldsymbol{\tau}=[\mathbf{u},\mathbf{y}]\sim\text{data}} \log p_\theta(\boldsymbol{\tau}|J, \mathbf{y}_0, \mathbf{y}_T) \tag{4}$$

where the dynamics constraint $\Psi(\mathbf{u}, \mathbf{y}) = 0$ is implicitly learned from the data.

We construct the aforementioned generative model $p_\theta$ using a conditional diffusion process:

$$q(\boldsymbol{\tau}^k|\boldsymbol{\tau}^{k-1}), p_\theta(\boldsymbol{\tau}^{k-1}|\boldsymbol{\tau}^k, [J, \mathbf{y}_0, \mathbf{y}_T]) \tag{5}$$

**Conditional Optimization Using Classifier-Free Guidance:** During training, we input conditioning values indicating the optimization goal $J$, enabling the model to discern the quality of trajectories while learning control trajectory dynamics constraints. For each training trajectory, we calculate and normalize a label $r(\boldsymbol{\tau}) \in [0, 1]$ proportional to the optimization objective $J$. To generate optimal trajectories that deviate from the random training dataset, we input the label $r(\boldsymbol{\tau}) = 0$ into the conditional generative model and employ classifier-free guidance with weighted sampling, sacrificing diversity for optimality.

**Targeting Using Inpainting:** Given the initial state and desired target state in our problem, modeling whether the generated trajectory accurately reaches the endpoint within the aforementioned optimization objective and including it in the condition label is approachable. However, for such explicit and discrete targets, we adopt a more direct method: we not only input it as an additional condition to the diffusion denoising network but also treat it as an inpainting problem similar to image generation, utilizing repaint techniques to ensure generation quality (Lugmayr et al., 2022). The repaint technique improves the consistency of generated content through a resampling denoising strategy.

## 4.2 Denoising Network Design and Inverse Dynamics

**Denoising Network Design:** We parameterize the denoiser using a dual-Unet framework consisting of two consecutive 1-D U-nets. 1-D Unet is a combination of multiple 1-D convolutional layers. The dimension of $\boldsymbol{\tau}$ ($M + N$) corresponds to the channel dimension of the CNN. The U-Net effectively captures the relationship between control signals and system observation states along physical time dimention, thus better modeling the complete trajectory distribution (Janner et al., 2022). Condition labels $r$ and diffusion time steps $k$ are encoded through MLPs as $\mathbf{r}_{\text{emb}}$ and $\mathbf{k}_{\text{emb}}$, respectively, and concatenated as input to the U-Net. For initial and target states $\mathbf{y}_0$ and $\mathbf{y}_f$ condition, we found they significantly decide the optimal trajectory/control signal distribution. Therefore, we use them to explicitly adapt the network's output at positions close to the output.

Yan et al. (2012) suggests that for linear systems, the optimal control signal has a linear relationship with $\mathbf{y}_c = \mathbf{y}_f - A\mathbf{y}_0$ (a linear combination of initial and target states, A is related to the network's parameter). We consider this linear operation in our model as the first-order expansion of optimal control with respect to $\mathbf{y}_c$ for nonlinear systems, with coefficients learned through the first U-Net. Correspondingly, we attempt to add another Unet to learn the coefficients of higher-order (second-order) expansions at the back end, fine-tuning the first-order results through residual connections. We first learn $\mathbf{y}_c$ by inputting $\mathbf{y}_0$ and $\mathbf{y}_f$ through a single linear layer without bias. The network formula is expressed as follows:

$$\mathbf{C}_1 = \text{Unet}_1(\boldsymbol{\tau}^k, [\mathbf{k}_{\text{emb}}, \mathbf{r}_{\text{emb}}]), \qquad \mathbb{R}^{B \times T \times C_1}, \mathbb{R}^{B \times C_2} \to \mathbb{R}^{B \times T \times (C_1 \times N)}, \tag{6}$$

$$\mathbf{O}_1 = \text{reshape}(\mathbf{C}_1) \cdot \mathbf{y}_c, \qquad \mathbb{R}^{B \times T \times (C_1 \times N)}, \mathbb{R}^{B \times N} \to \mathbb{R}^{B \times T \times C_1}, \tag{7}$$

$$\mathbf{C}_2 = \text{Unet}_2([\boldsymbol{\tau}^k, \mathbf{C}_1], [\mathbf{k}_{\text{emb}}, \mathbf{r}_{\text{emb}}]), \quad \mathbb{R}^{B \times T \times 2C_1}, \mathbb{R}^{B \times C_2} \to \mathbb{R}^{B \times T \times (N \times C_1 \times N)}, \tag{8}$$

$$\mathbf{O}_2 = \mathbf{y}_c^T \cdot \text{reshape}(\mathbf{C}_2) \cdot \mathbf{y}_c, \qquad \mathbb{R}^{B \times N}, \mathbb{R}^{B \times T \times (N \times C_1 \times N)}, \mathbb{R}^{B \times N} \to \mathbb{R}^{B \times T \times C_1}, \tag{9}$$

$$\hat{\boldsymbol{\tau}}^0 = \mathbf{O}_1 + \mathbf{O}_2, \tag{10}$$

where $B$ is the batch size, $T$ is the state sequence length, $C_1$ and $C_2$ are feature dimensions, and $N$ is the learned $\mathbf{y}_c$'s dimension. We illustrate the structural framework of the denoising network in Figure 1 and discuss its role in the experiments.

**Inverse Dynamics:** The diffusion model is used to capture the deep connections behind states, controls, and constraints, considering both the relationship between control and state and learning the dynamics behind state evolution. However, in real-world scenarios, most control and observed state associations are relatively simple. For common time-invariant systems, control linearly intervenes in state evolution, and only the current time-adjacent state and current action are relevant. However, the complex diffusion model with a global perspective may overfit this relationship, complicating simple associations and ultimately making state prediction difficult to generalize. Additionally, actions are less smooth than states, making their distribution more challenging to model. Therefore, inspired by Agrawal et al. (2016); Ajay et al. (2022), rather than directly sampling both control signals and intermediate states using the denoising network, we update the prediction of control (action) sequence by inputting the generated state trajectory to an inverse dynamic model $f_\phi$: $\mathbf{u}_{t,\text{update}}^0 = f_\phi(\mathbf{y}_t^0, \mathbf{y}_{t+1}^0)$. We use an Autoregressive MLP as the model and optimize it simultaneously with the denoiser via training data. Our final optimization loss function is:

$$L(\theta, \phi) := \mathbb{E}_{k, \boldsymbol{\tau} \in \text{data}, \beta \sim \text{Bern}(p)}[||\boldsymbol{\tau} - \boldsymbol{\tau}_\theta(\boldsymbol{\tau}^k, (1 - \beta)r(\boldsymbol{\tau}) + \beta\emptyset, k, \mathbf{y}_0, \mathbf{y}_f)||^2] \tag{11}$$

$$+ \mathbb{E}_{(\mathbf{y}_t, \mathbf{u}_t, \mathbf{y}_{t+1}) \in \text{data}}[||\mathbf{u}_t - f_\phi(\mathbf{y}_t, \mathbf{y}_{t+1})||^2] \tag{12}$$

where $\text{Bern}(p)$ is a binary distribution used for unconditional training.

### 4.3 RETRAINING

Randomly generated training data cannot guarantee coverage of optimal scenarios. To generate near-optimal controls that may deviate significantly from the training distribution, we need to guide the model in generating out-of-distribution data. On one hand, classifier-free guidance can make the generated distribution deviate from the original distribution. Therefore, we can use the data initially generated by the model, which has already deviated from the training distribution and tends towards the optimal distribution, to retrain the model and expand its exploration space. This approach is predicated on ensuring the quality of generated data, at least guaranteeing that the samples generated by the model conform to the underlying dynamics.

Consequently, we extract the control sequence part from the generated samples (i.e., the output of inverse dynamics $\mathbf{u}_{\text{update}}^0$) and reintroduce it into the system to interact and generate corresponding observation sequences $\mathbf{y}_{\text{update}}^0$. Together we add the renewed $\tilde{\boldsymbol{\tau}}^0 = [\mathbf{u}_{\text{update}}^0, \mathbf{y}_{\text{update}}^0]$ to the retrain data pool used for fine-tuning. Note that we still do not need to obtain system parameters here. We also discuss the effect of this method in our experiments.

## 5 EXPERIMENTS

Our experimental design aims to address three key research questions: (1) Can DiffCon demonstrate superiority over current finite-time optimal control methods for complex systems control? (2) Can DiffCon be generalized to practical cases where the task could be more challenging and out-of-distribution of training data? (3) Do the proposed designs help DiffCon achieve better performance?

### 5.1 DATASETS

For our experiments, we prepared two complex system datasets. The data in both sets were collected by randomly initializing the system states and injecting control sequences to observe the resulting system behavior. Our main experiments were conducted in two representative scenarios: a ring system governed by the **Kuramoto** dynamics model (Acebrón et al., 2005), described by first-order ordinary differential equations, and the New England power-grid network system, governed by the **Swing** dynamics model, described by second-order ordinary differential equations (Susuki et al., 2011).

The Kuramoto model, as an abstract and classical model, is widely applied to various synchronization phenomena, such as biological rhythms and brain networks (Tang et al., 2014). It is known for its ability to capture the essence of synchronization in coupled oscillators, making it suitable for studying emergent collective behavior. The Swing model, on the other hand, is a fundamental representation of power system dynamics, crucial for understanding stability and control in electrical grids. It incorporates more physical constraints and interactions typical in power systems, including inertia and damping effects (Susuki et al., 2011). These two models exhibit different characteristics in terms of complexity and nonlinearity. The Kuramoto model offers insights into abstract synchronization problems, while the Swing model provides a more concrete, application-oriented scenario. Both models present a balance of challenge and representativeness, making them ideal for our study.

We collected data by randomly generating initial states and control signals from Gaussian distributions. The resulting dataset was then partitioned into training and test sets for the experiments presented in Table 2, thus evaluating the in-distribution performance of each method. Note that these data points are not necessarily optimal. For a detailed description of the experimental setup, please refer to the appendix.

### 5.2 BASELINE METHODS

We compared our approach with two state-of-the-art model-free finite-time optimal control methods. The first is proposed by Baggio et al. (2021), an analytical approach that leverages data to determine the optimal input for steady-state control of complex networks, without requiring knowledge of the underlying dynamics. The second is DiffPhyCon, introduced by Wei et al. (2024), which harnesses generative diffusion models to directly optimize system trajectories and control sequences over the entire horizon.

Figure 2: 8-point ring network

| Dataset | Kuramoto | | Swing | |
|---|---|---|---|---|
| | Target Loss | Energy | Target Loss | Energy |
| DiffPhyCon | 1.79E-02 | 1.108 | 0.110 | 0.167 |
| Baggio et al. (2021) | 1.56E-05 | 1.016 | 0.034 | 0.095 |
| Ours | **1.14E-05** | **0.737** | **0.006** | **0.080** |

Table 1: Performance comparison on the case study of the Kuramoto dynamics. $m1$ and $m2$ denote the winding number of the starting and ending states, respectively. Here we investigate two different cases.

| Case | (m1,m2)=(4,2) | | (m1,m2)=(2,0) | |
|---|---|---|---|---|
| | Target Loss | Stable Time | Target Loss | Stable Time |
| DiffPhyCon | 8.55 | – | 3.39 | – |
| Baggio et al. (2021) | 1.25 | 870 | 1.10 | 830 |
| Ours | **0.0872** | **210** | **0.0244** | **509** |

Table 2: Comparison of Target Loss and Energy across different datasets and models

These model-free baselines not only represent the current state-of-the-art in this problem setting but also share a crucial characteristic with our method: they do not rely on environment interaction during testing. Instead, they can generate optimal control strategies directly given the specified conditions. Detailed descriptions of these baseline methods can be found in the related works section.

### 5.3 Main Performance and Case study

**Kuramoto Dynamics.** For the Kuramoto model, we consider a simple but insightful example of a ring network of N=8 Kuramoto oscillators. The dynamic of the phases (states) of oscillators can be expressed by:

$$\dot{\theta}_{i,t} = \omega + sin(\theta_{i-1,t-1} - \theta_{i,t-1}) + sin(\theta_{i+1,t-1} - \theta_{i,t-1}) + u_{i,t-1}, i = 1, 2, ..., N. \quad (13)$$

The network system can to two kinds of stable equilibria, namely synchronous state $\tilde{\theta}_{i,t} = \omega t$ and splay state $\hat{\theta}_{m,i,t} = \omega t + \frac{2\pi * m * i}{N} + c$, where $c$ is a constant and $m$ is an integer denoting the winding number. Such a case is representative of the ring network structure, and the sinusoidal coupling function is fundamental in studying synchronization phenomena across various disciplines, from neuroscience to power systems. The presence of both synchronous and splay states demonstrates the rich dynamical behavior of even simple Kuramoto networks.

In the main experiment presented in Table 2, we report the performance of our method and baselines on the test set after applying the training data. We evaluate two objectives from the optimization function in Equation 1: **Target Loss** $L(\mathbf{y}(T), \mathbf{y}_f)$ and **Energy** $J(\mathbf{u})$. The results demonstrate that our proposed method outperforms the baselines, achieving reductions of 26.9% and 27.4% in the two metrics, respectively, compared to the second-best method. This indicates that our approach effectively captures the control and state dynamics constraints within the data. Furthermore, our method exhibits superior performance for nonlinear systems compared to the approach proposed by Baggio et al. (2021), probably due to our design of the denoising network for nonlinear systems and the introduction of inverse dynamics. DiffPhyCon appears to be the worst, showing that it faces more challenges in modeling complex non-linearity and control distribution than our method.

We also conducted case studies of steering tasks to explore the effectiveness of steering the system from a splay state ($m = 2$) to a synchronous state ($m = 0$), and from one splay state ($m = 4$) to another ($m = 2$). These tasks fall outside the distribution of the training set generated by random perturbations, allowing us to investigate the out-of-distribution generalization capabilities of various methods. To assess whether each method accurately guides the state to the target, we evaluate not only the Target Loss but also introduce a **Stable Time** metric. This metric is defined as the time at which the norm of the state change between consecutive time steps falls below 1e-8, with the control start time set as 0 and the control time step set to $T = 16$. The Kuramoto dynamics and

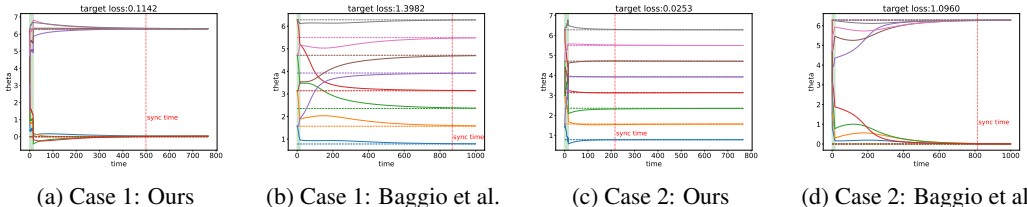

(a) Case 1: Ours  (b) Case 1: Baggio et al.  (c) Case 2: Ours  (d) Case 2: Baggio et al.

Figure 3: Visualization of the steering performance of our method and Baggio et al. for the two cases

ring network system ensure that when the final state can be controlled within the attraction range of the desired equilibrium, the network will gradually achieve the required synchronization pattern. Table 1 presents the results of our method compared to the best-performing method from Baggio et al. (2021) in the main experiment. Figure 3 visualizes the control effects of each method. Both the metrics and visualizations demonstrate that our method can most accurately transfer the state within a finite time, requiring only 24% and 61% of the duration needed by the second-best method to guide the state into a steady state for the two cases, respectively. Compared to Baggio et al. (2021), we achieve better performance because of the superior modeling capabilities of our model for non-linear dynamics. DiffPhyCon fails to steer the states, reflecting our method's stronger out-of-distribution generalization ability and practical value. These are likely attributable to our denoising network's ability to adapt its output to conditional inputs in real-time, and the retraining method that helps guide the model to explore out-of-distribution solutions.

**Swing Dynamics.**We also tested the control performance of various methods in the context of swing dynamics governed by second-order ordinary differential equations. The background of the system is the New England power grid network, which comprises 29 load nodes and 10 generator nodes. Using the first generator as a reference, the electromechanical behavior of the other 9 generators can be modeled by the swing equations, which reveal the dynamic constraints of generator phase and frequency. In this setup, the state is defined to include both the phase and frequency states of the 9 generators, resulting in a more complex state distribution. Our control signals are directly applied to the right-hand side of the phase differential equation, directly influencing the change in phase state. We provide a detailed description of the dynamics and experimental setup in the appendix.

Table 1 presents the performance of each method on the test set. Our method continues to demonstrate superior performance, achieving reductions of 82.4% and 15.8% in the two metrics, respectively, compared to the second-best method. The ranking of methods regarding energy consumption and control accuracy remains consistent with the Kuramoto scenario. Therefore, we can draw similar conclusions, namely that our method can better model complex relationships even when faced with nonlinear dynamics involving higher-order time derivatives.

This consistent performance across different dynamic systems underscores the robustness and versatility of our approach. The significant improvements observed in both the Kuramoto and swing dynamics scenarios suggest that our method's architecture, particularly the adaptive denoising network and the incorporation of inverse dynamics, provides a more comprehensive framework for capturing and controlling complex nonlinear systems.

## 5.4 ABLATION STUDY

To elucidate the contributions of different components in our proposed method, we conducted an ablation study on both the Kuramoto and Swing dynamics systems. Figure 4 presents the results of this study, comparing the Target Loss and Energy consumption for four variants of our model: our full model(**Ours**), without resampling (**w/o sampling**), without second-order adaption mentioned in section 4.2 (**w/o 2nd order**), and without both first and second-order adaption (**w/o 1st&2nd order**).

The results reveal consistent patterns across both systems, with our full model achieving the lowest Target Loss and Energy consumption. Removing the resampling component led to noticeable performance degradation, particularly in the more complex Swing system, underscoring its crucial role

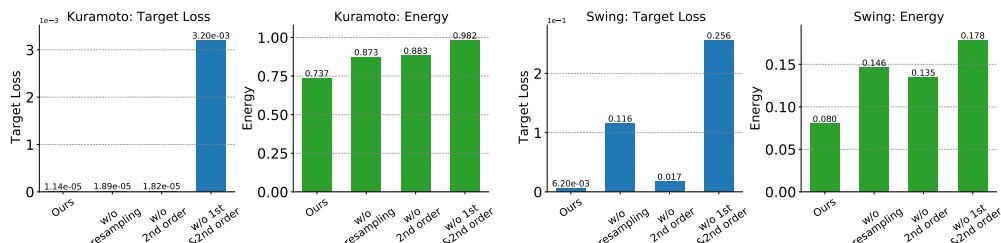

Figure 4: Ablation study on both the Kuramoto and Swing dynamics systems.

in handling intricate dynamics. The absence of second-order adaption has a moderate impact on performance, with its effect being more pronounced in the Swing system for Target Loss. Notably, removing both first and second-order adaptions results in the most significant performance drop in both systems, highlighting the synergistic effect of these components. The varying impacts observed between the Kuramoto and Swing systems indicate that our method adapts differently to systems of varying complexity. These findings emphasize the importance of each component in our method: resampling is vital for maintaining low Target Loss and Energy consumption, and the combination of first and second-order dynamics is essential for achieving optimal performance across different dynamical systems.

# 6 CONCLUSION

In this paper, we proposed DIFOCON, a novel diffusion model-based framework for finite-time optimal control in complex systems with nonlinear dynamics. By reformulating the control problem as a generative task, DIFOCON leverages a dual-Unet architecture to capture nonlinear dynamics and generate complete control sequences without relying on system models or parameters. Our approach integrates an inverse dynamics module and a retraining method to enhance performance on complex systems and improve out-of-distribution generalization. Experiments on two typical nonlinear complex systems demonstrate DIFOCON's superior performance, achieving significant reductions in target loss and control energy while exhibiting faster convergence than existing model-free approaches. These results underscore DIFOCON's effectiveness as a data-driven solution for finite-time optimal control in complex, nonlinear systems where traditional model-based methods fall short. This work shows the potential of data-driven generative models in addressing finite-time optimal control challenges in complex systems where accurate models are unavailable.

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

## A  EXPERIMENT SETUP

To evaluate the performance of DIFOCON, we conducted experiments on two representative complex systems: the Kuramoto model and the Swing dynamics model. These systems were chosen for their ability to demonstrate rich dynamical behaviors and their relevance to various real-world applications.

### A.1  KURAMOTO DYNAMICS

The Kuramoto model is a paradigmatic system for studying synchronization phenomena. We considered a ring network of $N = 8$ Kuramoto oscillators. The dynamics of the phases (states) of oscillators are expressed by:

$$\dot{\theta}_{i,t} = \omega + \sin(\theta_{i-1,t-1} - \theta_{i,t-1}) + \sin(\theta_{i+1,t-1} - \theta_{i,t-1}) + u_{i,t-1}, i = 1, 2, ..., N. \quad (14)$$

This system can converge to two kinds of stable equilibria:

1. Synchronous state: $\tilde{\theta}_{i,t} = \omega t$

2. Splay state: $\hat{\theta}_{m,i,t} = \omega t + \frac{2\pi m * i}{N} + c$, where $c$ is a constant and $m$ is an integer denoting the winding number.

For the Kuramoto model, we generated 20,000 samples for training and 1,000 samples for testing. The initial phases were sampled from a Gaussian distribution $\mathcal{N}(0, I)$, and the random intervention control signals were sampled from $\mathcal{N}(0, 0.1I)$. The system was simulated for $T = 16$ time steps with $\omega = 0$. The resulting phase observations and control signals were used as the training and test datasets.

## A.2 Swing Dynamics

The Swing dynamics model, which is crucial for power system stability analysis, is described by the following equations:

$$\dot{\delta}_i = \omega_i, +u_i \tag{15}$$

$$\frac{H_i}{\pi f_b} \dot{\omega}_i = -D_i \omega_i + P_{mi} - G_{ii} E_i^2 + \sum_{j=1, j \neq i}^{10} E_i E_j (G_{ij} \cos(\delta_i - \delta_j) + B_{ij} \sin(\delta_i - \delta_j)). \tag{16}$$

Here, $\delta_i$ is the angular position or phase of the rotor in generator $i$ with respect to generator 1, and $\omega_i$ is the deviation of the rotor speed or frequency in generator $i$ relative to the nominal angular frequency $2\pi f_b$. The parameters $H_i$ and $D_i$ are the inertia constant and damping coefficient, respectively, of generator $i$. $G_{ii}$ is the internal conductance of generator $i$, and $G_{ij} + iB_{ij}$ (where $i$ is the imaginary unit) is the transfer impedance between generators $i$ and $j$. $P_{mi}$ denotes the mechanical input power of generator $i$, and $E_i$ denotes its internal voltage.

For the Swing dynamics model, we generated 20,000 samples for training and 1,000 samples for testing. The initial phases were sampled from a Gaussian distribution $\mathcal{N}(0, 0.5I)$, and the random intervention control signals were sampled from $\mathcal{N}(0, 0.01I)$. The system was simulated for $T = 32$ time steps with $\omega = 0$. The resulting phase observations and control signals were used as the training and test datasets.

Both datasets provide a diverse range of initial conditions and control inputs, allowing for a comprehensive evaluation of DIFOCON's performance in handling complex, nonlinear dynamical systems.

