# OpenReview forum: "A Diffusion-based Generative Approach for Model-free Finite-time Control of Complex Systems"
_ICLR.cc/2025/Conference — Submitted to ICLR 2025_

### Official Review · Reviewer_Nf31 · 2024-10-29

**Soundness:** 2
**Presentation:** 2
**Contribution:** 3
**Rating:** 5
**Confidence:** 2

**Summary:**

This paper introduces a finite-time optimal control framework to guide systems to target states in the case of unavailable dynamic model. A dual-Unet is desigend to capture nonlinear system dynamics and generate entire control sequences in a single step. In an attempt to enhance generalization performance, a retraining strategy is added.

**Strengths:**

The presented framework is relatively complete, including to a diffusion model with a dual-Unet architecture, an inverse dynamics module, and a retraining strategy.

**Weaknesses:**

The readability of this paper should be improved. The presence of measurement noise does not seem to be considered in the two simulation tests, which is unreasonable. Moreover, the nonlinearity of both simulation tests is weak.

**Questions:**

1)	The author claims that the optimization framework contains a denoiser part. Please explain the concrete principle of reducing noise. Moreover, the measurement noises are not reflected in both experiments. Provide more details.
2)	The power consumption and detailed network framework of the presented method should be shown in all tests.
3)	The specific function of inverse dynamics module in experiments cannot be fully elucidated. Ablation study is required.
4)	These experiments are insufficient to show the advantages of the proposed framework. Both ring model and swing model can not fully represent a real physical system. I wonder if this method is effective for highly nonlinear dynamic systems such as n-Dof manipulators, ground vehicles ans quadcopters? Please enrich relevant practical examples.
5)	The readability of this paper should be improved. Some linguistic mistakes can be found in context.

---

### Official Review · Reviewer_3t2Z · 2024-11-01

**Soundness:** 3
**Presentation:** 3
**Contribution:** 2
**Rating:** 5
**Confidence:** 4

**Summary:**

The paper studies the problem of computing open-loop control sequences to solve an optimal control problem for a system with nonlinear, unknown dynamics. The approach is based on diffusion algorithms, particularly sampling both state and control signals from a diffusion model.

**Strengths:**

The paper addresses a challenging control problem (optimal control of nonlinear systems) in a challenging setting (system dynamics are unknown). The use of diffusion algorithms to solve controls problem is also intriguing and novel, to the best of my knowledge.

**Weaknesses:**

Unfortunately, the paper doesn't offer performance guarantees, which is a major drawback of any controls method. Does the computed control sequence solve the optimal control problem or not? If not, what is the performance gap? Without guarantees, it would be difficult to recommend the method for any practical control applications.

The (numerical) performance of the proposed method is also not convincing. Sure, the method may work better than other alternatives is some cases, but without formal guarantees it is difficult to speculate that this will be the case for other systems as well. Then, when should the proposed method be used? Additionally, one of the existing methods used for comparison is not designed for nonlinear systems, making it expected that the proposed method may have better performance in these cases. Perhaps, methods based on the Koopman operator or recent techniques based on feedback-linearization (or other data-driven methods for nonlinear control) should be used to really validate the performance of the proposed methods.

**Questions:**

What kind of performance guarantees does the proposed method have?

How does the method perform compared to data-driven methods for nonlinear systems?

How does the proposed method perform in a larger class of dynamical systems?

---

### Official Review · Reviewer_WVzx · 2024-11-02

**Soundness:** 2
**Presentation:** 3
**Contribution:** 2
**Rating:** 3
**Confidence:** 4

**Summary:**

This paper proposes a conditional diffusion model for solving optimal control problems with nonlinear dynamics. This method is purely data-driven without using information from the system dynamics. The authors also propose a dual-Unet architecture and learnable inverse dynamics module to help improve the performance of the diffusion models. In addition, retraining are used to help address the distribution shift issue. The method is tested on optimal control problems with two nonlinear dynamics, and showing improvement in target loss and energy.

**Strengths:**

This paper is well organized and the writing is easy to follow. The experiment part conducts an ablation study of the proposed building blocks.

**Weaknesses:**

1. The novelty of this paper is unclear. There are many papers that use the conditional diffusion model to generate optimal control solutions in a purely data-driven fashion. For example, there is no discussion on what is the key difference between this paper and [1], and even [1] also uses an inverse dynamics module.
2. The baseline selection process is not clear. For example, why does the author choose DiffPhyCon as there are other diffusion methods like [1] that can be used in similar tasks? Not to mention that in Table 2, the DiffPhyCon doesn't have a stable time. Overall, the diffusion model results are far behind other methods. What is the intuition behind it?
3. The experiment results are not very convincing. In Table 1, Table 2, and Figure 4, no statistics are provided for the results.  The difference in the results can be marginal.


[1] Anurag Ajay, Yilun Du, Abhi Gupta, Joshua Tenenbaum, Tommi Jaakkola, and Pulkit Agrawal. Is conditional generative modeling all you need for decision-making? arXiv preprint arXiv:2211.15657, 2022.

**Questions:**

1. For the related work, since you aim to generate a trajectory sequence, why not discuss existing work on open-loop control?
2. From line 216, can you talk more about how you treat targeting using inpainting with more details?
3. During the data retraining, when you use the data generated by the model, even though it might be feasible, it might not have a good objective/energy. How do you ensure that this data will help training?
4. Do you use target loss and energy as conditional input to the diffusion model?

---

### Official Review · Reviewer_iUx3 · 2024-11-04

**Soundness:** 2
**Presentation:** 2
**Contribution:** 2
**Rating:** 3
**Confidence:** 3

**Summary:**

This paper presents a diffusion-based approach which casts the model-free finite-time control problem of physical systems as a generative task. The provided methodology includes some helpful additions such as utilizing a dual-Unet architecture, an inverse dynamics module and retraining for enhancing performance. Experimental results on two systems show the advantages of the proposed approach against two baseline methods.

**Strengths:**

The strengths of this submission are outlined as follows:

1. This paper provides an interesting interpretation of control problems as generative tasks shown in Eq. (4), (5).

2. The authors provide useful modifications in their methodology such as a dual U-Net architecture, the incorporation of an inverse dynamics module as well as iterative training scheme for enhancing exploration.

3. The experiments indicate that the proposed framework outperforms the two provided baselines for both studied dynamics models.

4. A useful ablation study is presented highlighting the advantages of each added component in this work.

**Weaknesses:**

The main weaknesses/limitations of this paper can be summarized as follows:


1. The theoretical contribution of this paper appears to be limited. In particular,

    a) The main theoretical contribution is the introduction of problem (4) which is a standard log likelihood maximization. Furthermore, it is not well justified why the selection of this optimization is proper for the control of complex systems. The authors should better justify how it relates to the original problem (1) and whether the conditioning in Eq. (4) occurs based on an underlying derivation or if it is ad-hoc approach.

    b) The classifier-free guidance free idea has also been presented in a related approach in [R1]. To the reviewer's best understanding, the difference between the current paper's approach and [R1] is only on how labeling works.


2. The related work section is short and only emphasizing in few works, rather than providing a general overview of the areas. For example, in Section 2.1, a large body of literature on deep learning based control is omitted. In addition, only two references are provided for finite-time control methods. The authors are encouraged to provide a more complete overview of the related literature, as this is of great importance for the reader to understand the motivation and importance of a proposed method.

3. It is unclear whether a running state cost or constraints can be incorporated through the proposed formulation, although such specifications are often crucial to be met in complex physical systems. The problem formulation in Eq. (1) only includes a terminal state cost, and similarly, in Eq. (4) the conditioning is only on the initial and terminal states.

4. It seems that in Eq. (4) there is also a conditioning on the "optimization goal" J which is the desired cost. Nevertheless, such an approach might encounter the following limitations: i) It is often very hard to "predict" what a good cost is - especially in complex physical systems. ii) If the cost J used for conditioning is worse (higher) than the optimal cost of Eq. (1), then the proposed approach might "force" the resulting policy to be worse than it should. On the other hand, if the guess for the optimization goal is too good to be feasible (too low), then no trajectories will satisfy this conditioning. The authors are encouraged to comment on this issue.

5. The actual implementation of the proposed methodology is not clearly explained in the paper. An algorithm figure is missing showing the steps and how the described components are integrated in practice (e.g., inpainting, inverse dynamics).

6. The advantages of the presented method are only shown in two systems. The authors are encouraged to explore more complex physical systems with performance specifications encoded throughout the tasks, constraints, etc.

[R1] Li, A., Ding, Z., Dieng, A. B., & Beeson, R. (2024). Efficient and Guaranteed-Safe Non-Convex Trajectory Optimization with Constrained Diffusion Model. arXiv preprint arXiv:2403.05571.

**Questions:**

1. The authors are encouraged to elaborate on the derivation of Eq. (4), (5) and how it is connected to the original problem in Eq. (1). Is there an underlying mathematical derivation that is missing or is the proposed formulation in Eq. (4), (5) an ad-hoc approach for tackling problem (1)?

2. Can the proposed methodology be extended for handling running state costs and/or state constraints? Given that the motivation for this approach is the control of complex physical systems, these are specifications that are often significant to be met in practice in such applications.

3. To the reviewer's best understanding, conditioning on the optimization goal J in Eq. (4) can lead to issues such as the ones described in weakness (4). Could the authors provide a clarification on that?

4. Although the inpainting idea sounds interesting, it is not clearly explained and especially how it applies on a control of complex systems perspective. The authors are invited to further elaborate on how inpainting works in their problem setting.

5. While the experiments section is helpful to evaluate the performance of the method, its scalability is not investigated. How does this framework scale with an increasing dimensionality in the studied problems?

6. In Eq. (11), the authors use the variable $\tau$ to refer to the dataset consisting of the set of trajectories and controls. However, it appears that in the optimization $\tau$ also corresponds to the parameterized conditional classifier-free diffusion model. Could the authors provide a clarification on why the same symbol is used for both?

---

### Meta-Review · Area_Chair_UyjK · 2024-12-16

**Metareview:**

The reviewers ratings poses a formulation of optimal control as a form of  maximum likelihood optimization. While many reviewers found the formulation though-provoking, most all struggled to see the novelty of the contributions of the work, especially given the rich body of existing work in the generative model control space. The authors declined to include a rebuttal, and reviewers were all below an acceptance threshold. Hence, this is a clear case of rejection.

**Additional Comments On Reviewer Discussion:**

Given the unanimous low reviews and absence of author rebuttals, no discussion took place.

---

### Decision · Program_Chairs · 2025-01-22

Reject